# Environmental flows or economic woes—Hydropower under global energy market changes

**Moritz Schillinger[1], Hannes Weigt[1], Philipp Emanuel Hirsch [2]***

**1** Research Centre for Sustainable Energy and Water Supply, University of Basel, Basel, Switzerland, **2** Department of Environmental Sciences, Program Man-Society-Environment, University of Basel, Basel, Switzerland

* philipp.hirsch@unibas.ch

**Data Availability Statement:** This EPEX SPOT Market Data is owned by a third party organization. Other researchers may obtain the data in the same way as the authors by accessing the data at https://data.open-power-system-data.org/time_series/

## Abstract

The global energy system changes towards renewables-dominated and liberalized markets. This requires making novel trade-offs between the profitable development of hydropower and its environmental effects on the natural flow regime. Here, we used a pristine river as a model for how these future changes will affect the natural flow regime and identify future changes on previously overlooked levels. We found that damming and discharging based on market prices leads to first- and second-level deviation from natural flows. Beyond these effects, we identified a third level of distance from natural flow. This third level is created by the transition towards a renewables-dominated energy system. The volatile energy input from renewables incentivizes hydropower plant operators to discharge based on more flexible trading behavior. We conclude that novel economic models be combined with tailored implementations of environmental flows. This will allow to find novel solutions for the trade-off between market liberalization and sustainable hydropower development.

## Introduction

### Hydropower and sustainable development

As the world accelerates towards a new energy system dominated by carbon-free renewable energies, novel challenges emerge. These challenges need to be tackled to ensure a sustainable development of our energy supply. Hydropower plays a special role here, because it is a cornerstone of our current global electricity provision and its future development will be essential to meet the Sustainable Development Goal (SDG) 7 'Affordable and clean energy' [1]. Hydropower usage always leads to trade-offs between energy supply and environmental protection. The anthropogenic discharge dynamics are a major threat to the functioning of natural rivers and wetlands [2] (Fig 1). This functioning relies on the dynamic nature of a natural flow regime [3, 4]. Functioning rivers and wetlands provide essential ecosystem services and are central in achieving the SDGs [5]. Already today, about half of the global river volume is impacted by fragmentation and flow regulation due to hydropower usage [6]. With the global

(for the years 2002 to 2015) or by sending their requests to EPEX SPOT SE, Market Data Department (contact via marketdata. technical@epexspot.com, or by phone at +33 1 73 03 61 81).

**Funding:** The work was funded by the Swiss Commission for Technology and Innovation (CTI) under Grant No. KTI. 1155000154. The funders had no role in study design, data collection and analysis, decision to publish, or preparation of the manuscript.

**Competing interests:** The authors have declared that no competing interests exist.

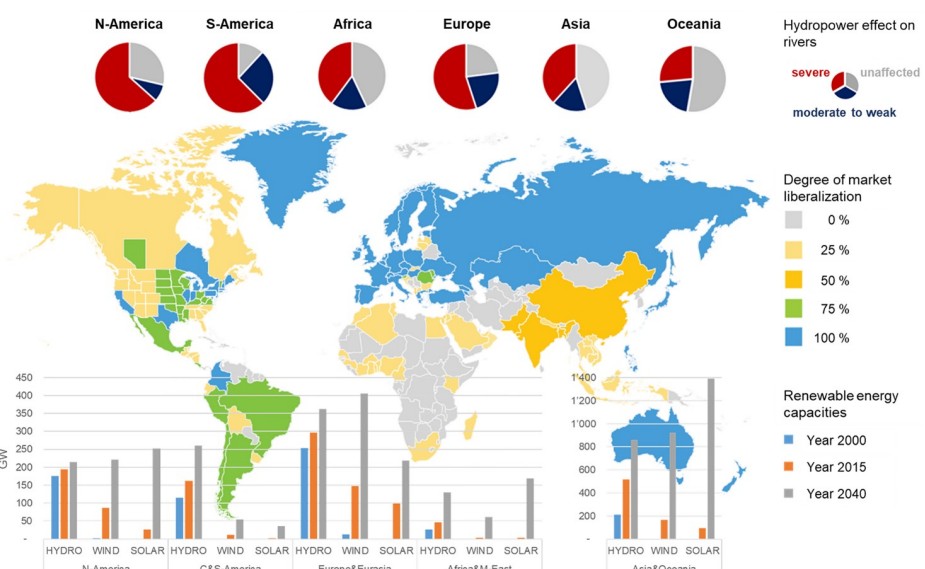

**Fig 1. Global impact of hydropower on river systems, electricity market structures, and renewable capacity development.** A degree of market liberalization of 0% means full regulation while 100% means full competition. Sources: Global flow regime threat from ref. [6]; electricity market structures based on ref. [10]; renewable capacity from ref. [7, 8]. World map in the background reprinted from "IEA. Re-powering Markets: Market design and regulation during the transition to low-carbon power systems: Electricity Market Series. 2016. All rights reserved." [10] under a CC BY license, with permission from IEA, original copyright 2016. Disclaimer: This map is without prejudice to the status of or sovereignty over any territory, to the delimitation of international frontiers and boundaries and to the name of any territory, city or area.

share of undeveloped hydropower remaining high—especially in Asia, Africa, and South America [7–9]—the installed hydropower capacity is expected to significantly increase in the coming decades (Fig 1): The IEA World Energy Outlook [8] assumes a total hydropower investment volume of $1.7 trillion until 2040. This enormous future development is likely to intensify the conflicts between achieving SDG 7 and the SDGs aiming to ensure ecosystem services and biodiversity (i.e. SDG 6 'Clean Water and Sanitation', 14 'Life Below Water', 15 'Life on Land'). A recent study [9] quantified these trade-offs: Globally an unexploited hydropower potential of 5.67 PWh yr$^{-1}$ was estimated. When imposing ecological flow restrictions to protect the natural flow regime (by diverting 30% of the discharge through regulatory mandated flows), this potential is almost halved. This conflict highlights a second, scientifically under-appreciated dimension of conflict: between markets and sustainable development.

## New markets, new challenges

The global energy system change will re-shape the way energy is provided and traded. The two major trends of this re-shaping are market restructuring and renewables generation [11]. In the last two decades many electricity systems around the globe were restructured; formerly regulated and monopolized structures were replaced by market competition (i.e. European and American electricity markets, see Fig 1). New actors entered those systems and put pressure on existing structures. Similarly, renewable generation has experienced a significant increase in the last two decades and solar and wind capacities are expected to dominate the future system (e.g. ref. [12, 13], see also Fig 1 for projections up to 2040). Higher shares of wind and solar generation require more flexible plant operation, because their intermittent nature makes electricity production more volatile. In concert, these two developments

highlight the role and need for more flexible market structures and point to a gap in our knowledge on the possible environmental ramifications of these developments. Economic research identified the first reactions to these new markets: the emergence of increased short term trading in recent years (i.e. up to 5 minutes before delivery [14]) and new market products such as capacity mechanisms were implemented to ensure sufficient supply in case of low wind and solar injection [15]. Hydropower will be the key element for adapting to these markets by providing the needed flexibility. Unlike wind and solar plants, the operators of storage hydropower plants are able to hold back water and decide when to discharge, thus providing the crucial ability to balance supply and demand [16, 17].

These developments of the energy system have ramifications on hydropower, which have yet to be fully acknowledged by researchers. For example, hydrological research has greatly advanced our understanding of how hydropower influences the hydrological regime [18]. Ecological research has greatly advanced our understanding on how the hydrological regime affects the environmental parameters, such as fish spawning habitats. However, interdisciplinary research including the economic sciences is an urgently needed further advancement. Such research is needed to identify the impacts that the novel market developments could have on the flow regime of rivers [19]. The results of such research will then allow to account for processes that go beyond the models, that have traditionally informed environmental regulations.

## The three levels of hydropower impacts on natural flows

The conflicts between the SDGs on top of the novel market developments call for integrated scientific approaches. We will need scenarios and tools for decision making on how to sustainably develop hydropower, whilst consolidating possible conflicts and trade-offs [20]. This decision making is especially relevant for the legislative framework, that will require a scientific basis for review of authorisations and extensions of permits for hydropower. For example, the European Water Framework governs whether potential hydropower developments proceed and determines their modes of operations [21]. Similar legislation exists elsewhere in the world [22]. This legislation often requires protection of endangered or threatened species, protection of rare habitats, and sometimes consultation with local communities [23]. All of these additional aspects relate to the extent to which the flow regime is affected by hydropower generation. This emphasizes the need for a detailed understanding of how the future market developments will influence the main goal of hydropower operation—generating revenue for the operators.

Against this backdrop, we will provide a first analysis whether and how the electricity market of the future will affect river flows. In detail, we will analyze if this adds additional challenges for environmental flow protection, the achievement of the SDGs, and the development of market structures. We first aim to structure the impact of hydropower on flow regimes and the subsequent environmental conditions and incorporate the above-described global system changes into a traceable framework. To that end, we identify first, second, and third-level deviations of anthropogenic flows from the natural flow regime: The first level reflects the fact that, regardless of the system conditions, any hydropower installation (that is not a purely run-of-river system) will lead to alteration of the natural flow regime due to damming and storing water for later usage. Even in a stable electricity system, i.e. the regulated 'old energy system', this will lead to challenges for environmental protection. On the second level, hydropower will lead to a new flow dynamic as discharge becomes governed by anthropogenic market drivers. Acknowledging these new market drivers will be especially relevant in newly developing installations: new installations frequently have to pass through stringent environmental legislation

and planning constraints. The resulting flow controls imposed on the operator are chiefly based on traditional hydrological and economic models (e.g. [9]).

With the emergence of renewable energies the market dynamics will be shaped less by conventional generation and their underlying seasonal and yearly dynamic, but more by highly fluctuating supply of intermittent generators. Finally, in the 'new energy system' with the dynamics imposed by the ongoing market restructuring and increasing intermittent generation, we anticipate a third level of deviation from natural flows. This third level reflects a novel market era. In this era, hydropower operators will react to multiple potential drivers affecting their goal to maximize their revenue by going from a simple single-market focus (aiming at energy provision) to a multi-market focus (aiming at providing both energy and flexibility). This market superimposition will create a novel anthropogenic forcing on top of the traditional deviations from natural flows. Whilst both the second and third levels hinge on anthropogenic drivers, we differentiate here between traditional single-market and future multi-markets.

## A Central European case study exemplifying the global change

The complexity and diversity of hydropower installations and electricity systems around the globe is boundless. To conceptually analyze the future developments, we therefore need a well-described energy system and environmental background knowledge, such as high-resolution run-off data describing a natural river flow. We found these conditions in a virtual case-study approach in a real-world energy and environment setting: we use the German electricity markets and a well-researched alpine river. Germany can be seen as a testbed for global future developments: it rapidly expands its share of renewables, while at the same time restructuring the market environment [11, 24] (see Fig 2). Consequently, the conditions prevailing in recent years in Germany are a looking glass into the future challenges that global electricity systems will face. We couple this market background with flow data from the alpine river Sense as the representative system for natural river conditions. The Sense is a 36 km long river with 9 m$^3$/s mean run-off and one of the few remaining pristine alpine rivers in Central Europe without any anthropogenic influences [25, 35]. Being very well monitored, it will provide the needed

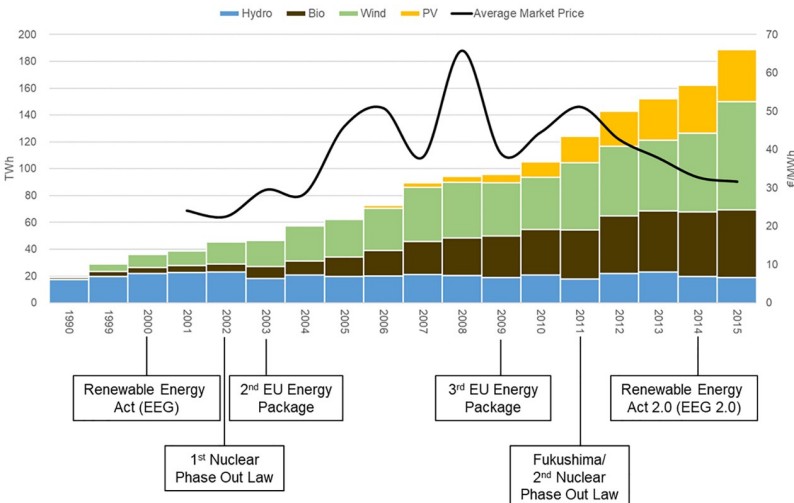

**Fig 2. Historic development of generation from renewable energies, average market prices, and market reforms in Germany.** Bio = energy from biomass, PV = photovoltaic energy. Sources: Development of renewable energies from ref. [30]; market prices from ref. [31]; and market reforms based on ref. [32].

reference data (daily discharge data from ref. [36], specifications in the Methods section) for the flow assessments. In terms of its size, the Sense stands representative for the global surge in the development of small hydropower plants (SHP, mostly less than 10 MW, although no strict definition exists). A recent global synthesis found that 11 SHP are currently installed for every one large hydropower plant [26]. The global number of 82,891 SHPs is expected to triple and countries like Brazil, China, and Africa are likely to see a leveling off in large hydropower projects, whereas SHP development will surge [26–28]. For our analysis, we apply a model representation of hydropower stations developed in ref. [29] and tailor it to the run-off data for the Sense (see the Methods section for details). The relevant outputs are quantifiable targets for both the natural flow regime and the hydropower operation for maximizing revenue. Following the three-level framework, we implement a storage hydropower station with a dammed river segment and then let it operate according to different market drivers. Firstly, we replicate average system conditions, reflecting a situation dominated by either relatively stable regulated structures or markets dominated by mostly conventional fossil power plants. This will allow us to identify the basic alteration embedded in damming a river (Level 1). Secondly, we analyze how the virtual Sense hydropower plant would have operated in the last decades on the German market. This will allow us to get an indication for how market restructuring efforts and new price dynamics imposed by wind and solar may alter flow conditions (Level 2). Third, we will extend the trading behavior of the hydropower plant by including system service provision as alternative income besides pure energy sales (Level 3). This multi-market strategy represents a more progressive setting, in which not only the generation but also the flexibility of the hydro station is used as an asset. This will allow us to analyze the impact of increasingly volatile market conditions on flow conditions in the 'new energy system'.

## Material and methods

### Hydropower operation model

The hydropower operation model used in this paper is similar to the model described in ref. [33] or ref. [17]. In the model, the hydropower plant operator maximizes the total revenue $R$ given by the revenue on the respective market ($m$) the plant is active on. When having a look at the Level 1 and 2 deviations from the natural flow, the hydropower plant is only active on the electricity market (day-ahead, DA). However, when taking into account trading imposed variations (Level 3), the hydropower plant provides additional services to the electricity system: the ability to balance supply and demand by discharging or holding back water (also called "balancing").

$$\max R = \sum_m R_m \tag{1}$$

In the DA market, energy is traded the day before actual delivery, with hourly products being exchanged on an hourly basis. The revenue on the DA market is the sum of the hourly DA market prices $p_{t,DA}$ and the generation $G_{t,DA}$ on the DA market over a period of one year.

$$R_{DA} = \sum_t p_{t,DA} G_{t,DA} \tag{2}$$

Due to uncertainties or technical problems, imbalances between demand and supply can occur after electricity (day-ahead and intraday) market clearance. However, since demand and supply have to be balanced at all times, the Transmission System Operators (TSOs) tender different balancing products to compensate changes in frequency. In this paper only one of these balancing products, namely secondary control reserve (SCR), which is tendered on the SCR

market, is considered to take into account potential trading imposed variations in the flow (see e.g. ref. [14] for more details on balancing markets). In order for the TSO to be able to compensate for imbalances in the electricity system at any time, SCR suppliers must reserve their generation capacity for the duration of the contract. This reservation of generation capacity as well as the actual production of electricity are remunerated in the SCR market. The capacity $Cap_{t,SCR}^{+/-}$ which is sold in the SCR market is remunerated by the capacity price $p_{t,SCR}^{cap+/-}$ and the actual generation $G_{t,SCR}^{+/-}$ is remunerated by the energy price $p_{t,SCR}^{energy+/-}$. In this paper, the German SCR market is taken into account. In the German SCR market weekly asymmetric products are traded. Thus, suppliers can bid separately for the provision of positive and negative balancing capacity (see ref. [33] for details). If a hydropower plants bids for positive SCR, it needs to increase its generation at times when additional energy is required to balance the electricity system $G_{t,SCR}^{+}$. If the plant is active on the negative SCR market, it needs to decrease its generation in times during which the electricity system is oversupplied $G_{t,SCR}^{-}$. The weekly structure of the SCR market implies a weekly contract period (all hours of a week are contracted).

$$R_{SCR} = \sum_t p_{t,SCR}^{cap+/-} Cap_{t,SCR}^{+/-} + \sum_t p_{t,SCR}^{energy+/-} G_{t,SCR}^{+/-} \tag{3}$$

The total generation or capacity is constrained by the hydropower plant capacity $cap^{max}$. If the plant is participating in the upward (positive) SCR market, $Cap_{t,SCR}^{+}$ has to remain free in order to be able to increase generation by the offered capacity level. To participate in the downward (negative) SCR market the hydropower plant needs to run on the DA market at the minimum capacity level $cap^{min}$ plus the capacity which was bid into the negative SCR market $Cap_{t,SCR}^{-}$ in order to be able to decrease its generation at times when the electricity system is oversupplied. In our case, the minimum capacity $cap^{min}$ is assumed to be zero (see below for details how downstream minimum flow requirements are considered). As mentioned above, the weekly structure of the SCR market translates into hourly restrictions for the power plant when participating in the SCR market. The capacity restrictions must therefore be fulfilled at all hours ($\forall t$).

$$G_{t,DA} + Cap_{t,SCR}^{+} \leq cap^{max} \quad \forall t \tag{4}$$

$$G_{t,DA} \geq cap^{min} + Cap_{t,SCR}^{-} \quad \forall t \tag{5}$$

The capacity bid into the SCR market can be called up completely or partly during the underlying period (here: week) by the TSO. Only what is called up ($call_{t,m}^{+/-}$) by the TSO has to be physically produced or reduced by the hydropower plant in the SCR market.

$$G_{t,m}^{+/-} = call_{t,m}^{+/-} Cap_{t,SCR}^{+/-} \quad \forall t, m = SCR \tag{6}$$

The generation or the reduction in generation is given by the water density $\rho$, the gravity $g$, the turbine efficiency $\eta$, the head $H$ and the discharge through the turbine $D_{t,m}^{+/-}$. In our case, the head is assumed to be constant (see further information on this condition at the end of this section).

$$G_{t,m}^{+/-} = \rho g \eta H D_{t,m}^{+/-} \quad \forall t, m \tag{7}$$

The actual discharge through the turbine $D_t^{net}$ at a specific point in time is the difference between the turbine discharge $D_{t,m}^{+}$ and the reduction in the turbine discharge $D_{t,m}^{-}$ in all

markets the plant is active on.

$$D_t^{net} = \sum_m D_{t,m}^+ - \sum_m D_{t,m}^- \quad \forall t \tag{8}$$

The turbine discharge can only be reduced by the amount which is discharged through the turbine at a specific point in time.

$$\sum_m D_{t,m}^- \leq \sum_m D_{t,m}^+ \quad \forall t \tag{9}$$

In addition, the actual turbine discharge is constrained by the maximum $d^{max}$ and minimum $d^{min}$ discharge capacity. The minimum turbine discharge is assumed to be zero. Downstream minimum flow requirements were assumed to be unusable for the power plant and have therefore already been deducted from inflows into the reservoir. Accordingly, the minimum flow requirements were considered in our approach as if this water would bypass the power plant.

$$d^{min} \leq D_t^{net} \leq d^{max} \quad \forall t \tag{10}$$

The storage level $S_t$ is defined by the storage of the previous time period $S_{t-1}$, the utilizable inflows $i_t$ (inflows minus the downstream minimum residual flow requirements), and the discharge. Water can be discharged either through the turbines ($D_t^{net}$) or it can bypass the turbines through spilling ($Spill_t$). However, spilling water would reduce the profit of a hydropower plant since water is not used to generate electricity. Accordingly, in the model, spill only occurs when the reservoir reaches its limits while the turbines are already running at full capacity (e.g. during floods). The case that spill may occur during extreme high flow events such as decade or century floods is not considered here. The total amount of water entering the downstream river is thus the sum of the turbine discharge and the spilled water. In addition, the minimum flow requirement previously deducted from the inflows is added.

$$S_t = S_{t-1} + i_t - D_t^{net} - S_{pillt} \quad \forall t \tag{11}$$

The storage is constrained by the maximum $s^{max}$ and minimum $s^{min}$ storage levels.

$$s^{min} \leq S_t \leq s^{max} \quad \forall t \tag{12}$$

The model is formulated as a linear program and coded in GAMS 24.7.4. The model is solved over a time period of one year using Cplex 12.6 while the time resolution is one hour (optimization over all hours of a year). Different contract time resolutions in the markets considered in this paper (DA and SCR) translate into hourly restrictions which are binding for the duration of the contracts. Each year that we consider in our analysis is solved individually, inter-annual dependencies are not considered endogenously. The model described above is subject to simplifications and assumptions that can have an impact on the obtainable quantitative results. One major drawback of the model formulation is its deterministic nature. The impact of uncertainty in the water inflows and the prices is neglected. Consequently, the resulting operational decision will represent the theoretically best benchmark given perfect information. Real operational decisions under uncertainty are likely to lead to less optimal behavior of hydropower plants. In addition to uncertainty, technical characteristics of hydropower, e.g. regarding head or turbine efficiency, are simplified in the model formulation. In reality, the head varies with the water level and influences the generation efficiency of the hydropower plant. However, the relationship between head (and water level) and efficiency is complex and site- and plant-specific. Since we use a generic hydropower plant in this paper (see next

section), we lack data on this relationship and we have refrained from generalizing this plant-specific relationship. Although the consideration of a non-constant head may slightly change the discharge of a hydropower plant, the most important criterion for when and how much water should be discharged, remains the market (price) and these are represented in high detail in our model. However, another limitation of our analysis is the fact that we take the perspective of a single hydropower plant and assume a perfect competitive market setting. While this is likely true for the electricity market, it may not necessarily hold for balancing markets [29].

### Virtual case study

In this paper, we rely on a virtual case-study approach. Therefore, a generic hydropower plant is combined with a case of a pristine river.

**Generic hydropower plant.** In ref. [29] three generic hydropower plants (large, medium, small) which are representative for Switzerland have been defined using the ratio of inflows to storage size and the ratio of storage capacity to turbine capacity as structural indicators. The structural indicators from ref. [29] of the large plant (ratio of inflows to storage size = 2; ratio of storage capacity to turbine capacity = 1000) have been used to define a generic hydropower plant representative for the case study river of this analysis. The derived generic hydropower plant considered in this paper can be classified as a seasonal storage hydropower plant with a storage capacity of 145 Mio. $m^3$ and a turbine capacity of 175 MW. It represents a typical alpine hydropower project in a high gradient system. While in this paper a generic hydropower plant is considered, the methodology can be applied to any other hydropower plant for which the required data are available. The application of a generic setup with stylized representations like in this paper can both under- and overestimate the real world counterfactuals, because hydropower plants usually have a high heterogeneity. Focusing on a generic setup therefore only provides an average benchmark when comparing the results to an individual real world plant. However, the model formulation is kept flexible to allow for the inclusion of more detailed plant specific data, if it is available [29].

**Case study river.** In this paper, the Swiss river Sense is used as an example for a pristine river. The Sense is one of the most natural river landscapes in Switzerland characterized by a variety of structures, a high consistency and natural fluvial dynamics. The Sense still has intact floodplains, which are classified in Switzerland as being 'of national importance'; its biodiversity includes i.e. the brown trout (*Salmo trutta*), a high species-richness of aquatic insects, as well as several amphibian and reptile species endangered in Switzerland [34]. The Sense emerges from the confluence of the "Warmer Sense" and the "Kalter Sense" in the canton of Fribourg. It forms the border river between the canton of Bern and the canton of Fribourg and enters the rivers Saane, Aare, Rhine and the North Sea. The Sense has a length of 36 km, a difference in altitude of 1100 m and a catchment area of 352 $km^2$ [35]. Daily run-off data for the Sense from 1993 to 2018 are available from the Swiss Federal Office for the Environment [36] while the average runoff is approx. 9 $m^3$/s. In this paper, the years 1993 to 2015 were chosen for the run-off data. In order to take the legal flow requirements in Switzerland into account, the run-off of the Sense, which can be used by the hydropower plant, is adjusted by subtracting the minimum residual flow. The concept of the residual flows has become contractual in Switzerland by an amendment to the constitution, the so-called Swiss Water Protection Act (WPA). The underlying rationale was to secure a minimum amount of flow that must not be used by hydropower but is meant to secure environmental functions of the river. It is calculated based on the mean $Q_{347}$ over 10 years (i.e. the average flow rate reached or exceeded on 347 days of a year) according to the WPA [37]. Depending on the $Q_{347}$ value, Article 31 in Section 2 of the WPA defines the minimum constant flow rate (e.g. for a $Q_{347}$ flow rate of 2500 l/s

the minimum flow rate is 900 l/s). We define $Q_{347}$ flow rate individually for each year we consider (1993-2015) instead of using a 10-years average. We do this, because we have no flow data before 1993, so we cannot use the 10-year average for the years before 2002. To be consistent, we also do this for the later years, for which a 10-year time series would be available. Using the year-specific $Q_{347}$ value instead of the 10-year average can lead to an over- or underestimation of the minimum flow requirements in a given year. Across all years considered, the average $Q_{347}$ value for our case study River Sense is approximately 2500 l/s (max: 3000, min: 1400), resulting in an average minimum flow requirement of 900 l/s (max: 1010, min: 570).

For additional details on the Sense see ref. [35] and ref. [34].

## Market scenarios

In this paper, the German electricity market is used as a case study. In order to compare the "traditional" hydropower operation with the operation of hydropower plants which is adapted to changes in the electricity system, the following two market scenarios are considered:

- single-market

- multi-market

In the "single-market" scenario, the hydropower plant is operated on the electricity (day-ahead, DA) market only. While in the "old energy system" (Level 1 impacts) electricity prices and dynamics were defined mostly by conventional technologies and stable market structures, the share of renewable energies significantly increased in the past years in the German market as market structures changed (see e.g. ref. [24] and Fig 2). In our simulations, we use the average hourly DA prices of the years 2001 to 2003 to represent the 'old energy system', while we consider the prices of the last 15 years (2001 to 2015) to analyze the dynamic changes on the German market (i.e. Level 2 impacts). The respective price years are coupled with the runoff data of the river Sense from 1993 to 2015. Thus, in the case of Level 1, an average price year (average of the years 2001 to 2003) and 23 runoff years (1993 to 2015) are combined. In the case of Level 2, a combination of 15 price and 23 runoff years is simulated (i.e. 345 combinations). In the "multi-market" scenario (i.e. Level 3 impacts) the hydropower plant is active on the electricity (day-ahead, DA) and balancing (secondary control reserve, SCR) market. Thus, the multi-market scenario takes into account trading behavior of hydropower by including alternative income in addition to pure energy sales. While there are multiple sources of alternative income, the SCR market is used as an example. Price data which are used in this paper are collected from two sources: hourly day-ahead market prices for Germany are based on ref. [31], SCR market prices are derived from ref. [38]. With regard to the SCR market, only the years 2012 to 2015 are considered (in terms of price and runoff data), as the model is computationally challenging when the SCR market is considered. Because the SCR market design in Germany changed in 2018, we consider the market design, which was in place before 2018 (see ref. [38] and ref. [33] for details). In the German SCR market, all participating suppliers submit a price quantity bid, while all suppliers that are accepted on the market are remunerated with their bid price (pay-as-bid auction). There is therefore not a single price on the market, but several. To derive a single price for our simulations, we take the average of all accepted prices, weighting the individual prices with the supplier's quantity. The energy price and the energy delivery (call up) are based on a merit-order (see ref. [14] and ref. [33] for details). To simulate the merit-order of the German SCR market, the energy price by supplier, the offered capacity and the called up of energy from ref. [38] are used. However, since we do not know the position of a plant or technology in the merit order, we assume that the hydropower plant operator

bids such an energy price into the SCR market that he/she is called up only if more than two-thirds of the total SCR capacity are required by the TSO.

## Sustainability boundaries and environmental flows

Given the desired increase in renewable generation and the respective need to finance new investments and maintain existing plants, we can expect that the novel market developments will increasingly affect hydropower development around the globe. The question then becomes whether flow regulations could revert this anthropogenic alteration and at the same time maintaining sufficient income streams to plant operators to finance their assets. To satisfy the need for specific targets in the development towards such a trade-off, ecologists developed the "Sustainability Boundary Approach" (SBA) as defined by ref. [39]. The SBA serves as a framework within which flow alterations are acceptable with the premise of securing the river's ecological integrity as much as possible. To facilitate the cooperation with water managers the SBA was conceived as a hydrologically oversimplified but tangible parameter, that can be specifically adopted to any given river. The SBA restricts the hydrological alterations to a range around the natural flow in order to facilitate a suitable water management. Ref. [40] propose that the range for hydrologic alteration should be 10% for a high level of ecological protection and 20% for a moderate level of protection. Thus, the daily flow should not increase or decrease more than 10 or 20% compared to the natural flow. It is important to note that these hydrological alterations are a presumptive standard, that the inventors intended for application if detailed scientific assessments of environmental flow needs cannot be undertaken in the near term. Such a situation however, is more than likely given the steep and fast increase in SHP, which we will see in the near future. In the analysis at hand, these proposed standards by ref. [40] are taken into account in order to analyze the trade-off between ecological protection and hydropower operation. The boundaries set by the SBA, the sustainability boundaries (SDB) are defined on the basis of the daily run-off of the respective year, which is simulated. This would require accurate run-off forecasts to define the SDBs in advance. If such forecasts are not available in reality, the SDBs could also be defined on the basis of historical average run-off data. While boundaries on the daily flow are in practice difficult to define [40], monthly flow boundaries are considered as well. To take into account the daily and monthly sustainability boundaries in the hydropower plant operation model, two additional constraints are added. The daily or monthly flow out of the reservoir, depending on whether daily or monthly SDBs are applied (both in this paper), given by the actual turbine discharge $D_t^{net}$ and the spill $Spill_t$, has to be within the predefined sustainability boundaries $SD^{min}$ and $SD^{max}$ (± 10 or 20% of the natural flow).

$$D_t^{net} + S_{pillt} \geq SD^{min} \quad \forall t \tag{13}$$

$$D_t^{net} + S_{pillt} \leq SD^{max} \quad \forall t \tag{14}$$

In addition, the minimum flow requirements under WPA are not taken into account in the analysis of SDBs, as the two requirements are not necessarily complementary. We therefore assume that the introduction of SDBs would replace the minimum flow regulation. When analyzing environmental flows (flood pulses, see below), only the minimum constraint from above is added to the model.

Beyond the generic SDBs as a presumptive standard set of flow deductions [40], the concept of environmental flows requires even more specific tailoring of discharge levels to any given river [41]. The environmental flows, that we consider here, were transferred and adapted from another river: the river Spöl. Based on data on a successful dam re-operation project in the

river Spöl, we modelled how such a dam re-operation would affect the revenue of our virtual hydropower plant at the river Sense. The river Spöl is an alpine river flowing in approximately 200 km distance from the Sense. The Spöl was dammed for hydropower. It has a very similar latitude, length, and catchment area than the Sense and both rivers' catchments have 0% glaciated area (see S1 Table for a detailed comparison of the rivers). The environmental flow design for the Spöl followed the concept of artificial peak flows and was implemented following a multi-year ecological study including experimental flows. These artificial flood pulses of high un-turbinated discharge were designed to flush out fine sediments and benthic algae. This ensures optimal spawning and hatching conditions for gravel-spawning salmonid fish like the brown trout, which are socio-ecologically important in many temperate rivers. Research showed that the implementation of the flows did indeed secure the spawning grounds and improve the salmonid's recruitment [42]. For the sake of our comparison, we assume the restrictions to be implemented at a river stretch that has a comparable mean run-off as the Sense. To address the trade-offs between securing ecologically relevant flows and economically profitable operation, we statistically tested how the generic SDB would affect the revenue under different scenarios, we used an ANCOVA model with revenue (€-1 MWh-1 week-1) as response variable, time (week) as co-variate, years as random factor, and two categorical predictors: on the one hand the two market scenarios (single and multi) and on the other hand no SDB vs. SDB. The resulting replicate number $n$ per test is 212 (53 weeks per each of the 4 years (2012-2015) for which the comparison of weekly revenues was calculated). As targets for the SDBs we use a 10% and 20% allowed deviation on daily and monthly flows [40]. This is a conservative estimate compared to the 30% which is commonly assumed as an ecological restriction [9]. The daily and monthly differentiation is meant to capture two diverse levels of restrictiveness on operational decisions. The daily restrictions represent a future in which environmental concerns have priority over economic considerations, whereas the monthly restrictions will allow more flexibility for the operator with arguably more profound effects on the river ecosystem.

While SDBs constrain the flows at all days or months of the year, the flood pulses are only considered at specific days of the year. The reference data for the flood pulses are based on the historic flood pulses in the river Spöl between 2012 and 2015. To translate the flood pulses of the Spöl to our case study river Sense, we consider the flooding level in the Spöl relative to the average flow conditions in the Spöl. For the timing of the flood pulses we use the same days for the floods as observed in the past in the Spöl (see ref. [36] for daily flows of the Spöl and Sense rivers).

## Results and discussion

### Level 1: Negating natural flows

Damming a river is by definition aimed at altering the natural flow pattern and subduing it to the needs of the electricity system. For our example case this is evident from comparing the natural flow variation over two decades with a virtual flow pattern when a dam is installed and operated according to 'old energy system' conditions (Fig 3). Evidently, when a dam is built the idiosyncratic natural flow dynamics are lost. We view this as the basic first-level impact of hydropower occurring in regulated electricity systems or markets mostly dominated by fossil and nuclear power plants that are characterized by rather stable operational conditions.

### Level 2: Imposing market driven flow variations

With the restructuring of electricity systems and the increasing share of renewable generation in many countries around the globe the formerly rather stable electricity system conditions are

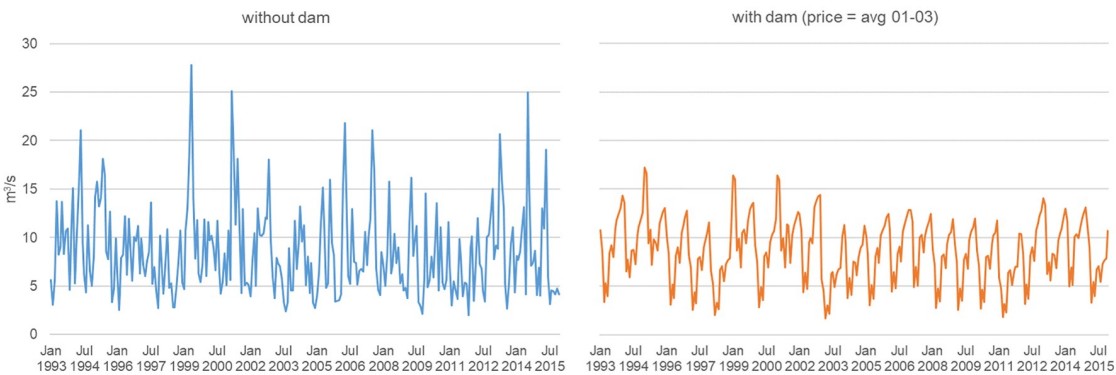

**Fig 3. Natural flow pattern vs. standardized flow pattern.** Average monthly flow volume for the period 1993 to 2015. The dam is operated according to market prices based on the average hourly prices of the German spot market from 2001 to 2003. A period during which the electricity market was dominated by fossil and nuclear energy leading to stable operational conditions.

increasingly replaced by more volatile market driven conditions. For an example, see historic developments in Germany (Fig 2). Those market conditions in turn influence the operation of hydropower plants and thereby the resulting flow pattern. We view these as the Level 2 impacts of hydropower operations, because flow alterations become operation-driven. For our example case this impact is evident from the comparison to the natural flow variation (Fig 4). Operation-driven flow alterations result in changes in seasonality and magnitude of flows and a in a different characteristic of the rising limbs. The seasonality and magnitude of low levels changes to even lower levels and different time points in the year than on Level 1, because the dam allows to greatly reduce flow volumes for long periods (to benefit from higher prices in later periods). The rises are less steep and more gradual, because they are governed by the turbine capacity in case of a dammed river. The exact pattern of deviation from Level 1 here is, of course, highly liable to different market drivers, whose development is uncertain in the future. The development of the 'new energy system' will depend on global fuel prices and carbon policies, the power plant mix of the country in question, renewable and further energy market policies, and electricity demand dynamics. To account for this dependency, we conducted several model runs with different future price scenarios, but there were no fundamental changes in the patterns of change (see S1 Fig).

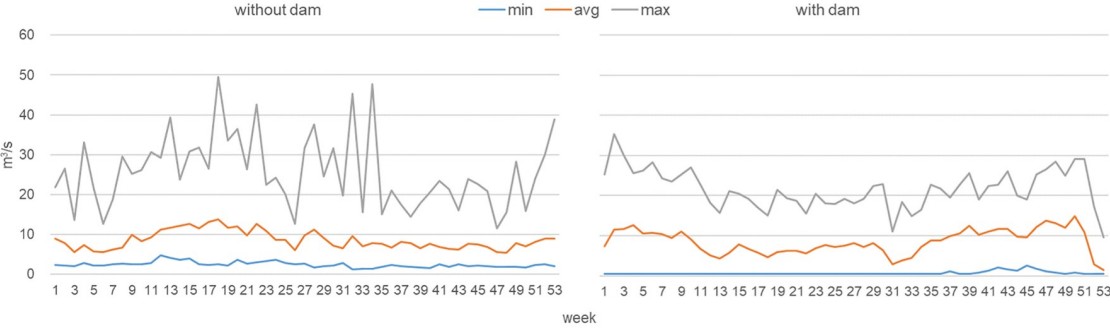

**Fig 4. Natural flow variation vs. market driven variation.** Based on average weekly flow volumes. The variation of the natural system represents the period 1993 to 2015. The variation of the market driven system is derived by combining the natural inflows from the period 1993 to 2015 with the market price variation in the period 2001 to 2015 (i.e. a matrix of 345 flow/price-year combinations).

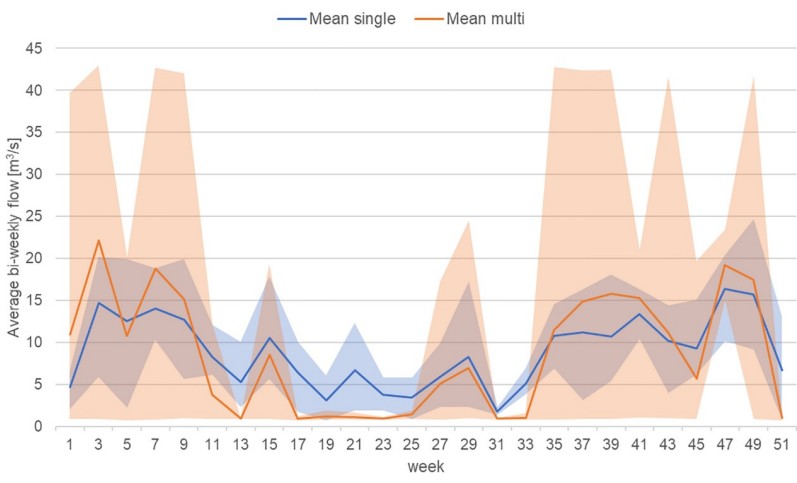

**Fig 5. Impact of trading activity on flow patterns.** Based on average bi-weekly flow volumes for the period 2012 to 2015. The single-market results represent a trading strategy focused on the hourly energy market, i.e. the 'old energy system'; the multi-market results represent a strategy allowing trading on energy and system service markets, i.e. the 'new energy system'. The shaded area represents the range of the bi-weekly flow volumes.

## Level 3: Adding trading-imposed variations

Finally, as hydropower plant operators adjust to the new markets in the 'new energy system' and optimize their trading structures to benefit from the high flexibility of hydropower, a new dimension of alterations on the river flow patterns is added. We view this as the Level 3 impact of hydropower introducing a novel level of anthropogenic alterations of flows. Fig 5 again shows this for our example case and the difference between a pure energy market oriented trading strategy (i.e. a single market) and an optimized trading behavior aiming at energy markets and system service provision (i.e. multi market in the 'new energy system'). The differences are pronounced with larger variations between peaking and low flow patterns in case of multi market activity. Importantly, the differences here apply in comparison to the level 2 changes in flow regime and not only in comparison to the level 1 or even the natural flow regime. This epitomizes the novelty of this level of change.

## Trade-offs between following markets and protecting the natural flow regime

Given the desired increase in renewable generation and the respective need to finance new investments and maintain existing plants, we can expect that the Level 3 flow alterations will increasingly become reality for river systems around the globe. Albeit the exact flow pattern will be highly river- and plant-specific, the three-leveled-structure should prevail in general. As the resulting flow patterns tend to lose their natural character, the question becomes whether flow regulations could revert this anthropogenic alteration and at the same time maintain sufficient income streams to plant operators to finance their assets. To satisfy the need for quantifiable targets in the development towards such a trade-off, ecologists developed so-called sustainability boundaries (SDB) and the concept of environmental flows [39, 41]. SDBs serve as a framework within which flow alterations are acceptable with the premise of securing the river's ecological integrity as much as possible. SDBs are allowable percentages of deviation from the natural magnitude of flow [39]. To facilitate the cooperation with water managers SDBs were conceived as a hydrologically oversimplified but tangible parameter that can be specifically adapted to any given river [39]. The environmental flows require even more

specific tailoring of discharge levels to any given river [41]. Based on ecological research a proxy for a specific ecosystem function of the river is identified and flows are implemented that secure this function [43]. For our analysis, we applied both generic SBDs and transferred a pre-existing tailored environmental flow regime from another Swiss alpine river, the Spöl, to the Sense. The rationale here was to use an elaborate quantification of environmental flows that can serve as an example for how the novel levels or flow alterations might impact sustainable hydropower development in the future.

Overall, our analyses showed, that the daily SDBs reduce revenues by up to 24% and by up to 7% with the more flexible monthly SDBs (ANCOVA Daily flow deductions: effect of SDB, F = 140.89, MS = 1.1811, df = 1, p = 0.0013; ANCOVA Monthly flow deductions: effect of SDB, F = 11.04, MS = 2.389, df = 1, p = 0.044, Fig 6). The importance of the market behavior is furthermore evidenced by the significant influence of the scenario (single vs. multi) on the revenue (ANCOVA Daily flow deductions: effect of 'scenario', F = 144.36, MS = 6.0611, df = 1, p = 0.0012; ANCOVA Monthly flow deductions, F = 55.74, MS = 2.1812, df = 1, p = 0.004). Naturally, with the more restrictive daily SDBs, the flexibility to discharge is so limited for the operator, that the actual market behavior does not make much of a difference in terms of flow patterns. Under the less stringent monthly SDBs, we observe more pronounced difference of the flow patterns between the different market scenarios. To test how specific environmental flow restrictions would impact the revenue, we compared the years 2012 to 2015 with and without artificial flood pulses (the pulses are based on the level and timing of the flood pulses in the river Spöl, see details in the Methods section). Overall, the artificial flood pulses affected the revenues to a lesser degree than the generic water deductions (Fig 6). The loss in revenue was largely independent of the actual timing of the artificial flood pulses, indicating some flexibility for the operator to further minimize the water that is lost for discharge through the turbines. This can be achieved by choosing flood times, that serve their ecological function but at the same time occur at times during which the operator is able to spend some water. These exemplary numbers have two major implications: Firstly, they highlight the challenge embedded in environmental flow protection in the increasingly volatile electricity market

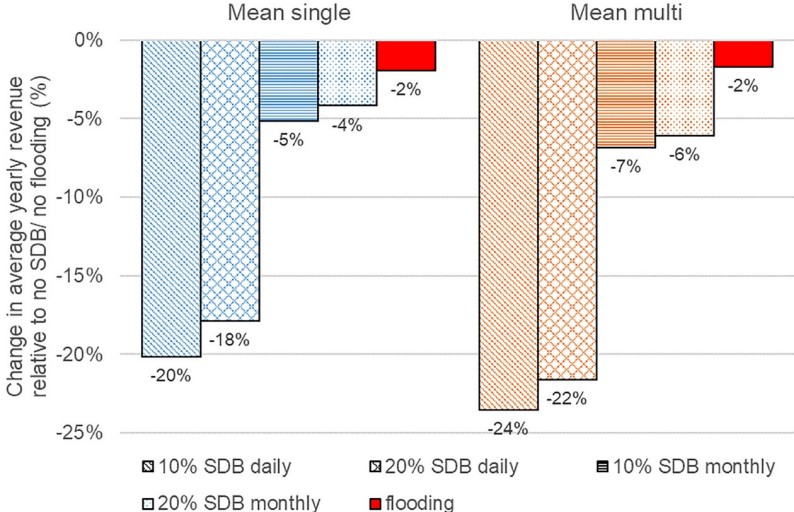

**Fig 6. Change in average yearly revenue by daily and monthly SDBs and environmental flows (flooding).** Based on yearly revenues for the period 2012 to 2015. SDBs represent 10% and 20% allowable deviations from the natural flow on daily or monthly basis. The environmental flows (flooding) are based on the magnitude, frequency, and timing of the environmental flows in the comparable river Spöl.

environment. The restrictions reduce the operator's flexibility to alter the natural flows. As flexibility will have an increasing value in the 'new energy system' dominated by intermittent renewables, these restrictions tend to increase the resulting economic impact compared to the 'old energy system' with relatively stable electricity system conditions. Secondly, they highlight that an intelligent design of environmental flows can minimize the loss in revenue, whilst potentially safeguarding an important ecological function of the river. On a global scale, an unspecific SDB of 30% ecological flow restriction has been estimated to cut the hydropower generation by almost half [9]. Based on such estimates and our analyses, it becomes clear that research towards tailored environmental flows is needed to optimize the trade-off between hydropower development and securing the natural river flow in the coming new market future.

Beyond the SDBs, there are more elaborate concepts that adopt a holistic stance towards the management of flow regimes by providing clear guidelines for good scientific practice (e.g. [44]). Our study complements these guidelines by highlighting the importance to include a novel energy system future into such research. The future management of flow regimes must appreciate that the market scenarios determine how flow management measures might affect the operators' revenue. It should also be noted here, that each environmental flow design is specific to a given river. For example, a research program would have to be instigated to specifically design environmental flows for protecting brown trout from adverse hydropower effects in the Sense. A growing body of literature provides guidelines for how environmental flows can be successfully implemented (recently synthesized in a review by [45]). All of the processes identified to be key to environmental flow implementation are affected by our findings. Firstly, legislation or policy towards environmental flows is essential to successful implementation: policies should be amended by including how much leeway is needed for the operator to be able to adapt to future energy system change. Secondly, an environmental impact study has to be conducted to identify limits and opportunities of flow regime management and also this impact study must acknowledge how the operators' decisions will be affected by a future energy system change. Finally, the necessary flow experimentation needs to be conducted to design environmental flows and even here the novel limits or opportunities on discharge that the energy system transition will bring, have to be acknowledged. At any case, stakeholders, researchers, and political decision makers must work together to identify an ecosystem service, establish a research program and prioritize the installation of environmental flows based on the research. For the river Spöl this is described in detail by ref. [46] and ref. [47] provide a conceptual treatise of such a transdisciplinary approach. Here, the novel implications are that economic modelling of future energy market developments should always be an integral part of such research approaches.

## Conclusion

### The challenge for a sustainable future global hydropower development

Our example is a shape of things to come on the global scale. The hydropower of tomorrow will lead to natural flows dynamics that are yet another level distant from natural flows. Although our testbed was located in Europe, the processes are relevant to the global hydropower development: with major dam projects, such as the Grand Ethiopian Reservoir Dam, receiving increasing backlash [48], there is a growing trend towards developing small, decentralized, carbon-free energy sources [26, 28]. Hydropower in rivers the size of our case-study will be prime targets for such future hydropower development. We acknowledge of course, that existing hydropower licenses already frequently have flow requirements that limit operational flexibility of hydropower plants to respond to the grid. These flow requirements may

limit ramp rates and discharges and thus the operator's flexibility to respond to the market. Our results here should therefore be put in the context of how such existing regulations will have to be or could be adapted in the novel energy systems of the future. To achieve both the desired increase in carbon free renewable generation (SDG 7) while maintaining the complex ecosystem services that rivers provide [9], environmental flow assessments will need to include more detailed electricity market representations to capture the resulting flow dynamics of hydropower operation. It is insufficient to use past market dynamics to predict future developments. Given the increasing dynamic of the energy market, a sustainable development of hydropower must consider the feedbacks between market decisions and environmental impacts. At the same time the sustainable hydropower development must account for both the revenue prospects of hydropower plant operators and the broader economic impacts of hydropower. Our analysis further demonstrates how a research-based approach of environmental flows might be a promising avenue for finding optimal solutions that safeguard a river's ecological integrity and minimize the losses to the hydropower operator's revenue. This highlights the need to find novel combined research approaches that enable an integrated economic and ecosystem development processes. These approaches must equip hydropower operators, environmental managers, and policy makers with river-specific forecasts of how the ecological integrity of a river can be protected by environmental flows that minimize the loss of revenue. Eventually, this will allow for more informed decisions on the optimal trade-off between market development and sustainable development.

## Supporting information

**S1 Fig. Monthly average flow without and with hydropower dam for different price scenarios.** Runoff values are based on the years 2004 and 2012; prices are based on the years 2015, 2030 and 2050. For the prices in 2030 and 2050, two future simulations (AFEM [49] and Linking [50] are considered.
(TIF)

**S1 Table. Comparison of basic characteristics of the rivers Sense and Spöl [35, 36, 51].**
(DOCX)

**S1 File. Data.** Data on hydropower plant, flows and balancing market.
(XLSX)

## Acknowledgments

This research was conducted within the SCCER CREST (Swiss Competence Center for Energy Research). We thank Dr. Rebecca Lordan-Perret, Joschka Wiegleb for commenting on earlier versions of the manuscript and especially Prof. Dr. Patricia Burkhardt-Holm for valuable inputs and support.

## Author Contributions

**Conceptualization:** Hannes Weigt, Philipp Emanuel Hirsch.

**Data curation:** Moritz Schillinger, Philipp Emanuel Hirsch.

**Methodology:** Moritz Schillinger, Hannes Weigt.

**Writing – original draft:** Hannes Weigt, Philipp Emanuel Hirsch.

**Writing – review & editing:** Moritz Schillinger.

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
