## [Decision Letter · Decision Letter 0]

20 May 2020

PONE-D-20-04709

Environmental flows or economic woes - hydropower under global energy market changes

PLOS ONE

Dear Dr. Hirsch,

Thank you for submitting your manuscript to PLOS ONE. After careful consideration, we feel that it has merit but does not fully meet PLOS ONE’s publication criteria as it currently stands. Therefore, we invite you to submit a revised version of the manuscript that addresses the points raised by the reviewer, all of which are important.

We would appreciate receiving your revised manuscript by Jul 04 2020 11:59PM. To enhance the reproducibility of your results, we recommend that if applicable you deposit your laboratory protocols in protocols.io, where a protocol can be assigned its own identifier (DOI) such that it can be cited independently in the future. For instructions see: http://journals.plos.org/plosone/s/submission-guidelines#loc-laboratory-protocols

We look forward to receiving your revised manuscript.

Kind regards,

Judi Hewitt

Academic Editor

PLOS ONE

Journal Requirements:

'This research is part of the activities of SCCER CREST (Swiss Competence Center for

Energy Research), which is financially supported by the Swiss Commission for

Technology and Innovation (CTI) under Grant No. KTI. 1155000154. We thank Dr.

Rebecca Lordan-Perret, Joschka Wiegleb and Prof. Dr. Patricia Burkhardt-Holm for

their valuable inputs and support.'

'The author(s) received no specific funding for this work.'

Additional Editor Comments (if provided):

Reviewers' comments:

Reviewer's Responses to Questions

**Comments to the Author**

1. Is the manuscript technically sound, and do the data support the conclusions?

Reviewer #1: Partly

2. Has the statistical analysis been performed appropriately and rigorously? 

Reviewer #1: N/A

3. Have the authors made all data underlying the findings in their manuscript fully available?

Reviewer #1: No

4. Is the manuscript presented in an intelligible fashion and written in standard English?

Reviewer #1: Yes

5. Review Comments to the Author

Reviewer #1: General comments

The general topic of this manuscript (ms) is of interest. The integration of an economic perspective to research into setting environmental flows would be a valuable and reasonably novel contribution to the literature. Unfortunately, I have five main areas of concern:

A. There are a several methodological clarifications required.

B. The structure is logical in terms of organisation of sections. However, several important methodological details are placed towards the end of the results and discussion section.

C. A broader appreciation of the breadth of approaches that have been adopted for environmental flow setting internationally is required through appropriate references to the literature.

D. The 3-level framework proposed is purely descriptive; and therefore subjective. This framework is devalued by the lack of a quantitative method.

E. Lack of any quantification of differences between downstream flow regimes is an unfortunate omission. There is a vast volume of literature on why particular flow regime metrics would be of interest to flow managers (e.g. minimum flows in specific months for fish spawning or passage).

Main issues

1) Level 1-3 is an interesting concept, but difficult for others to implement due to its narrative nature.

2) The presumptive standard of Richter et al. 2012 is applied. If possible, please give examples where this standard has been applied for a real hydropower operation. Apart from a sentence in the conclusion, there is no mention of legislative requirements/procedures (apart from Swiss minimum flow, and the ms does not explained how that is calculated) that would be applied to implement environmental flows in the real world. It is important to recognize that there has been great debate in the environmental flow setting literature in relation to flow setting methods. Approaches range between the “natural flow paradigm” and the “design-a-flow regime” points of view. See these references for examples discussing various approaches.

a. Acreman, M.C. and Ferguson, A.J.D., 2010. Environmental flows and the European water framework directive. Freshwater Biology, 55(1), pp.32-48.

b. Acreman, M., Aldrick, J., Binnie, C., Black, A., Cowx, I., Dawson, H., Dunbar, M., Extence, C., Hannaford, J., Harby, A. and Holmes, N., 2009, March. Environmental flows from dams: the water framework directive. In Proceedings of the Institution of Civil Engineers-Engineering Sustainability (Vol. 162, No. 1, pp. 13-22). Thomas Telford Ltd.

3) Lack of a quantification of differences between downstream flow regimes is an unfortunate omission. There is a vast volume of literature on why particular flow regime metrics would be of interest to flow managers (e.g. minimum flows in specific months for fish spawning or fish passage; higher flows for recreational kayaking at the weekend).

4) Hydro-peaking is given as a key word. The issue of hydro-peaking (rapid rises and falls in river flows that cause fish stranding) can be an important environmental consideration, but this topic is not tackled by the ms. Suggest removal of hydro-peaking from key words.

5) This is a multi-disciplinary study. Readers may therefore come from a variety of backgrounds. This situation calls for very careful use of technical terms that may me misinterpreted. Please see comments attached to pdf file for specific examples.

6) Please see comments attached to pdf file for numerous suggestions, comments and requests for clarification.

6. PLOS authors have the option to publish the peer review history of their article (what does this mean?). If published, this will include your full peer review and any attached files.

Reviewer #1: No

---

## [Author Response · Author response to Decision Letter 0]

6 Jul 2020

All responses are provided in the Response to Reviewers-file as requested.

---

## [Editor Report · Decision Letter 1]

14 Jul 2020

Environmental flows or economic woes - hydropower under global energy market changes

PONE-D-20-04709R1

Dear Dr. Hirsch,

We’re pleased to inform you that your manuscript has been judged scientifically suitable for publication and will be formally accepted for publication once it meets all outstanding technical requirements.

Kind regards,

Judi Hewitt

Academic Editor

PLOS ONE
---

## [Editor Report · Acceptance letter]

17 Jul 2020

PONE-D-20-04709R1 

Environmental flows or economic woes - hydropower under global energy market changes 

Dear Dr. Hirsch:

I'm pleased to inform you that your manuscript has been deemed suitable for publication in PLOS ONE. Congratulations! Your manuscript is now with our production department. 

Kind regards, 

on behalf of

Dr. Judi Hewitt 

Academic Editor

PLOS ONE